# Integrated Analysis of Cytokine Profiles in Malaria Patients Discloses Selective Upregulation of TGF-β1, β3, and IL-9 in Mild Clinical Presentation

**DOI:** 10.3390/ijms232012665

**Published:** 2022-10-21

**Authors:** Ella Larissa Ndoricyimpaye, Jacques Van Snick, Jean de Dieu Niyoyita, Philbert Kanimba, Jean Bosco Mbonimpa, Robert Rutayisire, Réverien Rutayisire, Vedaste Ndahindwa, Paméla Cheou, Jean Paul Coutelier, Nadine Rujeni

**Affiliations:** 1Biomedical Laboratory Sciences, School of Health Sciences, College of Medicine and Health Sciences, University of Rwanda, Kigali P.O. Box 3286, Rwanda; 2Medecine Expérimentale, de Duve Institute, Université Catholique de Louvain, 1348 Brussels, Belgium; 3Ludwig Institute for Cancer Research, Université Catholique de Louvain, 1348 Brussels, Belgium; 4National Reference Laboratory, Rwanda Biomedical Centre, Kigali P.O. Box 4285, Rwanda; 5School of Public Health, College of Medicine and Health Sciences, University of Rwanda, Kigali P.O. Box 3286, Rwanda

**Keywords:** malaria pattern, pro-inflammatory cytokines, parasite, regulatory cytokines, Tregs

## Abstract

The proper control of Plasmodium infection requires a finely balanced immune response. Here, we evaluated the implication of TGF-β1 and TGF-β3 in this process using novel monoclonal antibodies to measure their plasma concentrations in comparison with other cytokines and the expression of FOXP3 mRNA. Plasma cytokine levels were measured in 80 patients with severe anaemic malaria and 186 with a mild presentation using ELISA, and rtPCR was used to measure FOXP3 mRNA expression. While no mature TGF-β isoforms were detected in the plasma, the latent TGF-β1 and TGF-β3 were strongly upregulated in patients with mild malaria and nearly undetected in patients with severe disease. Similar selective upregulation in mild patients was observed for IL-9 and FOXP3 mRNA, while IL-7, IL-10, IL-17, and IL-27, although higher in mild cases, were also detected in severe disease. In contrast, a clearly skewed trend of severe cases towards higher pro-inflammatory (IL-6, IL-13, TNF-α) and Th1 (IFN-γ) responses was observed, which was associated with a higher level of parasitaemia as well as lower IgG and higher IgM responses. Together, these results suggest that the stimulation of regulatory T cells through TGF-β1/TGF-β3 and IL-9 is paramount to an effective and balanced protective immunity in natural human malaria infection.

## 1. Introduction

With more than 200 million cases and over 400,000 deaths per year, malaria remains a major World health concern [1]. *Plasmodium* parasites raise a complex challenge to the host immune system, requiring a strong Th1 immune response and strong antibody production to control parasitaemia but also for sufficient control of the response to avoid excessive inflammation, a common feature of severe malaria [2].

The human host–*Plasmodium* interaction leads to complex cellular responses, and TGF-β production seems to play a critical role as an essential moderator of malaria-induced inflammation (reviewed by Drewry [3]). In mouse malaria models, TGF-β production inversely correlates with disease severity [4] and modulates the production of inflammatory cytokines [5]. Moreover, it critically contributes to the development of FOXP3+ regulatory T cells after infection [6].

However, a causal relation between TGF-β and disease modulation is difficult to establish in natural human infection. The general trend is that TGF-β levels inversely correlate with disease severity [7,8,9] but positively with the increased numbers of FOXP3+/CD4+CD25+ T cells and rapid parasite growth [10,11]. However, an Indian study found higher levels of TGF-β in severely affected *P. falciparum*-infected individuals than in mildly affected patients and controls [12].

To further clarify the implication of TGF-β in malaria morbidity, we took advantage of novel anti-TGF-β antibody pairs to selectively measure TGF-β1 [13] and TGF-β3 [14] plasma concentrations in a panel of control and infected patients in Rwanda. Correlations were established with disease severity, FOXP3 expression, and a panel of 10 other cytokines. The ultimate goal was to determine the strongest determinants of disease severity and provide a general picture of pro- and anti-inflammatory cytokine expression in mild and severe malaria patients.

## 2. Results

### 2.1. Demographic Characteristics of Participants

A total of 324 participants were enrolled, 80 with severe anaemic malaria,186 with a mild clinical presentation, and a group of 58 non-infected controls (Appendix A). The mean age and sex ratio were comparable between the groups. However, the mean parasite density was significantly higher in severe cases than in mild cases (t = 12.356, *p* < 0.0001).

### 2.2. Identification of Human Plasmodium Infections Infecting the Participants

Using the reference procedure of standard microscopy, Plasmodium species were determined in our patients, and their prevalence rates in the two groups are illustrated in Table 1. To assess whether Plasmodium species could influence the pattern of cytokine distribution, we measured pro-inflammatory and anti-inflammatory cytokines’ expression in *P. falciparum*-infected and non-*P. falciparum*-infected patients separately. *P. falciparum*-infected and non-*P. falciparum*-infected patients exhibited the same trend of high concentration of regulatory cytokines (IL-9, TGF-β1, and TGF-β3) and low concentration of inflammatory cytokines (e.g., IL-6) in mild cases, while the opposite trend was observed in severe cases (high IL-6 and low IL-9 and TGF-β) (Appendix A).

### 2.3. Selective Upregulation of TGF-β1, TGF-β3, IL-9, and FOXP3 in Mild Malaria

Using a new set of anti-TGF-β1 and anti-TGF-β3 monoclonal antibodies that enable the specific and sensitive quantification of the activated forms of TGF-β1 and TGF-β3 (Appendix A), we failed to detect any active form of either factor (<20 pg/Ml). However, after the acid treatment of the plasma to liberate latent TGF-β from its latency-associated peptide, most of the patients with mild malaria showed increased levels of both TGF-β1 (1641 ± 945 pg/Ml) and TGF-β3 (5962 ± 2129 pg/Ml), compared with controls (TGF-β1: 174 ± 197 pg/Ml, TGF-β3: <20 pg/Ml) (*p* < 0.001 ANOVA). Conversely, in severe malaria, TGF-β1 was undetectable, and TGF-β3 was fourteen times lower than in mild cases (413 ± 6425 pg/Ml, *p* < 0.001) (Figure 1A and Appendix A).

IL-9 is a cytokine that is strongly upregulated by TGF-β [15,16], and, to the best of our knowledge, IL-9 plasma levels have not yet been studied and compared with TGF-β in malaria patients. Here, we observed that IL-9 plasma concentrations were comparable between controls (1676 ± 560 pg/Ml) and severe malaria patients (1905 ± 534 pg/Ml) but increased more than fourfold (6997 ± 1783 pg/Ml) in mild cases (*p* < 0.001) (Figure 1A), showing a striking similarity with TGF-β1 and TGF-β3.

Further analysis of the expression of TGF-β1, TGF-β3, and IL-9 in mild and severe individual patients showed highly significant positive correlations (Figure 1B). Remarkably, the correlation between IL-9 and TGF-β1 and TGF-β3 was as strong as that measured between the two TGF-β isoforms (Pearson r = 0.787 for TGF-β1–TGF-β3, 0.742 for IL-9–TGF-β1, and 0.758 for IL-9–TGF-β3).

The high plasma concentrations of latent TGF-β in mildly affected patients correlated with the selective and strong upregulation of FOXP3 mRNA expression (Figure 2), suggesting that biologically active TGF-β, although not detectable in our ELISAs, was present in these patients. Moreover, severely ill patients expressed lower FOXP3 mRNA than the control group. These results indicate a clear link between TGF-β1, TGF-β3, and FOXP3 expression in controlling the severity of malaria infection.

### 2.4. Multi-Factor Analysis in Mild and Severe Malaria

To put the above results in a broader perspective, factor analysis was performed including a panel of nine cytokines (IL-4, IL-6, IL-7, IL-10, IL-13, IL-17, IL-27, IFN-γ, and TNF-α) as well as IgG, IgM and patient characteristics (age, gender, parasitaemia, and haemoglobin concentration). The initial exploratory analysis showed no age and gender effect, and therefore, these were dropped from the analysis.

Cytokine and antibody measurements showed that all the cytokines and antibodies tested were upregulated by malaria infection but with clear demarcation between mild and severe cases. IL-6, IL-13, TNF-α, and IFN-γ were significantly higher in severe malaria, while IL-7, IL-10, IL-17, and IL-27 showed the opposite trend. IgG levels were higher also in mild cases, while IgM levels were higher in severe infections (Figure 3).

The subsequent dimensional analysis (Figure 4) indicated the strongest contribution of TNF-α, IFN-γ, and IL-13 towards disease severity, while parasitaemia had the weakest contribution. On the other hand, IL-10 had the least contribution towards a mild clinical outcome, while IL-9, TGF-β1, and TGF-β3 had the strongest. The levels of IL-4 were not significantly different between mild and severe cases.

### 2.5. Patterns of Cytokine Expression in Individual Patients

The analysis of the correlations between TGF-β1 and TGF-β3 was extended to other cytokines. The results showed a strong positive correlation between TGF-β1 and TGF-β3 with IL-7 and a weaker correlation with IL-17. There was also a positive trend for IL-27 and IL-10, but it remained below significance (Appendix A). In contrast, strong negative correlations were seen with IL-6, IFN-γ, and TNF-α, and no correlation was observed with IL-4 (Appendix A).

For better quantification of these correlations, we compared all log10-transformed cytokine concentrations using the Pearson protocol to obtain comparisons based on actual cytokine concentrations and group ranking, respectively. Strong, positive correlations between the distributions of TGF-β1 and TGF-β3, with high Pearson correlation coefficients, were observed (r = 0.863, *p* < 0.001). Similarly, significant, positive correlations were noted between IL-9 and both TGF-β1 (Pearson r = 0.898, *p* < 0.001) and TGF-β3 (Pearson r = 0.829, *p* < 0.001), demonstrating the coordination of the production of these factors in individual patients.

Another cytokine that showed strong positive correlations with both TGF-β1 and TGF-β3 was IL-7 (IL-7–TGF-β1 Pearson r = 0.612 *p* < 0.001 and IL-7–TGF-β3 Pearson r = 0.556 *p* < 0.001). On the other hand, there were significant, negative correlations of TGF-β1/TGF-β3/IL-9 with IL-6, IFN-γ, and TNF-α. A general pattern of cytokine correlations is shown in heat maps of Pearson r coefficients (Figure 5).

## 3. Discussion

Here, we present the cytokine levels and interactions from a cohort of malaria patients with different clinical presentations, as defined previously (i.e., severe anaemic malaria versus mild malaria) [17,18], aiming to understand anti-Plasmodium protective immunity. Using novel antibodies with selective specificity for TGF-β1 and TGF-β3, we observed the parallel upregulation of these two TGF-β isoforms in mild cases but not severe malaria infections. Notably, these new reagents detect free TGF-β1, not LAP-associated latent TGF-β, and no signal was seen without the acid treatment of plasma samples to liberate TGF-β from its latent form. This is in line with a previous report on the controlled *Plasmodium* infection of human volunteers, indicating that only a sharp peak of free TGF-β1 was detected and was limited to a few days (day 7–8) after infection before declining below detection by week 2 [10]. Since our patients are permanent residents of malaria-endemic areas, immunomodulation may explain the lack of detection of free TGF-β. Nevertheless, in the above-mentioned report, it was also shown that latent TGF-β1 concentrations perfectly correlated with the free form but were ±100-fold higher, suggesting that measuring latent TGF-β offers relevant information concerning TGF-β production during infection.

A strong, positive correlation was observed between TGF- β and IL-9, in line with the fact that TGF-β is known to stimulate IL-9 production by mouse T cells in vitro when combined with IL-4 [15,16]. The positive correlation that we observed suggests that the TGF-β–IL-9 connection operates in malaria patients. This could contribute to the observed upregulation of FOXP3 mRNA in mild malaria since TGF-β and IL-9 both contribute to Treg upregulation [19]. Since IL-9 also enhances the suppressive activity of FOXP3+ natural Tregs [20], a selective increase in TGF-β and IL-9 levels could provide an optimal environment for controlling excessive inflammation by stimulating Tregs. In addition, and similar to its critical role in the clearance of *Trichuris muris* [21], IL-9 could even enhance malaria parasite clearance by its ability to stimulate mast cells [22], which are essential for resistance to Plasmodium in the mouse [23]. This is consistent with the observation that mild malaria patients had a lower mean parasite density than patients with severe malaria in the current study.

Elevated IL-10 has been linked to asymptomatic malaria [24]. In our analysis, IL-10 was also increased in patients with mild malaria but, unlike TGF-β and IL-9, its concentration was also significantly higher in severe cases than in controls. This translated into much weaker correlations of IL-10 with TGF-β1, TGF- β3, and IL-9 than seen between the latter, thus making it a less reliable marker of malaria control.

Similar to IL-10, other regulatory cytokines such as IL-7, IL-17, and IL-27 were also increased in mild compared with severely ill patients, though the latter had higher titres than controls. Severe malarial anaemia has been linked to genetic variations in the IL-7 gene [25]. The upregulation of IL-7 in mild cases in our study suggests an important contribution of this factor to disease control rather than a response to erythropenia. Indeed, IL-7 could contribute to disease control given its implication in Treg biology [26,27]. In this context, it is relevant to mention the high correlation noted in individual patients of IL-7 with TGF-β1, TGF-β3, and IL-9, three factors implicated in Treg physiology.

A protective role of IL-27 against excessive inflammation during malaria infection has been suggested by the observation that IL-27-R-deficient WSX-1-/- mice show exacerbated disease [28,29]. Our data showed a significant increase in IL-27 only in patients with a mild disease which is consistent with a previous study on infants [30]. Notably, IL-27 promotes IL-10 production in *Plasmodium*-infected mice [31] and, in our patients, IL-10 and IL-27 expression patterns looked very similar, although their correlation did not reach statistical significance.

Another cytokine upregulated by TGF-β is IL-17 [32]. In our cohort of patients, IL-17 was increased more in mildly affected patients but still above the levels of controls in severely ill individuals where TGF-β was essentially absent. This suggests that, in malaria patients, other factors contribute to IL-17 upregulation. Moreover, discordant information has been published regarding IL-17 and malaria pathology [33,34]. Further mechanistic studies on host–parasite relationships will provide more insights into its role in disease modulation.

As expected, inflammatory cytokines such as IL-6, IFN-γ, and TNF-α were much higher in severe than in mild malaria patients. More surprising was the high concentration of Th2 cytokine IL-13. This cytokine is known for its inflammatory role in allergic and autoimmune disorders via mast cell activation and epithelial damage [35]. However, in humans, it can be produced by innate immune cells such as eosinophils, basophils, NK cells, mast cells, and NKT cells [36]. Its upregulation in severe malaria patients could be the consequence of the silencing of TGF-β1 and TGF-β3 and the associated drop in Tregs. It is, therefore, possible that its upregulation in the absence of regulatory cytokines and Tregs in malaria patients may lead to disease severity. Nevertheless, a protective role of IL-13 in malaria pathology was observed in individuals with a single C-to-T substitution in the IL-13 promoter [37].

It is worth noting that, in our study, *Plasmodium* species could not explain the cytokine-related differences between the two groups of patients. Indeed, there was no difference in immune responses between *Plasmodium falciparum*-infected participants and non-*Plasmodium falciparum*-infected cases, and the exclusion of patients with no falciparum infection did not change the outcome of our analyses. In addition, non-falciparum-infected patients exhibited the same cytokine pattern. This suggests that, regardless of *Plasmodium* species, the induction of regulatory cytokines in malaria infections induces protective immune responses, leading to the downregulation of pathogenic immune responses. These findings are consistent with observations in animal studies on different infectious diseases. Indeed, the treatment of *Plasmodium*-infected mice with neutralising antibodies to TGF-β exacerbated the malaria disease [4], while IL-9 played a protective role during experimental chronic infection with *T. cruzi* by reducing parasite multiplication and invasion [38]. Moreover, the treatment of mice with rlL9 was linked with decreased levels of IL-12, IL-6, and TNF-α levels in Chagas disease [39].

Taken together, our data disclosed a unique association between IL-9 and FOXP3-induced TGF-β1 and TGF-β3 that is a strong predictor for a mild clinical presentation in human natural malaria infection. This clinical malaria control is also associated with increased IL-7, IL-10, IL-17, and IL-27 plasma levels as well as IgG and better parasite clearance. On the other hand, the data confirm that severe disease is associated with the excessive production of pro-inflammatory cytokines that may also fail to induce antibody class switch that limits parasite density.

## 4. Materials and Methods

### 4.1. Study Population

Three hundred twenty-four (324) participants from three malaria-endemic provinces of Rwanda (South, East, and West) were enrolled. Of these, 266 were malaria positive, while 58 were healthy controls. The age range was 6 months to 56 years, and the sex ratio was 1:1.4 (190 males and 134 females). All the malaria patients in this study were recruited from healthcare centres, and blood was drawn before they were administered anti-malaria treatment. The 58 healthy individuals (control group) were randomly selected at the community level. The classification of each case of malaria was based on physical signs and symptoms, the laboratory results of malaria diagnosis, and the haemoglobin level at the time of presentation according to the World Health Organization (WHO) criteria [17]. No malaria patient exhibited cerebral malaria symptoms, and severe malaria categorisation was based on the presence of anaemia (defined as Hb concentrations < 5.0 g/dL or a haematocrit < 15.0 in children < 12 years old, and <7 g/dL or <20 haematocrits in adults). Blood haemoglobin was measured by using Hemocue (Hemocue AB), following the manufacturer’s instructions. The mild cases of malaria were defined as symptomatic malaria parasitaemia with no signs of severity and/or evidence of vital organ dysfunction.

### 4.2. Malaria Diagnosis

For the detection of parasitaemia, a calibrated thick-smear technique was used with standard Giemsa staining. Malaria status was confirmed independently by two trained laboratory technicians. Parasite densities were determined as (number of parasites counted/WBC counted) × WBC count/µL of blood, assuming an average of WBC of 8000 per microlitre of blood [40].

### 4.3. Cytokine Enzyme-Linked Immunosorbent Assay (ELISAs)

Blood samples were collected for cytokine assessment in sterile tubes containing EDTA. The samples were centrifuged, and the plasma was refrigerated at −80 Celsius degree until analysis. A quantitative enzyme-linked immunosorbent assay (ELISA) was performed according to the manufacturer’s protocol (Wuhan Fine Biotechnology Ltd., Wuhan, China) to determine the plasma levels of most cytokines and antibodies. TGF-β1 and IL-9 were analysed using homemade ELISA reagents. The capturing antibodies (MTGFb12C11J1481 and MH9A4J113) and biotinylated antibodies (MTGFB18D6/H511 and MH9A3F215) for TGF-β1 and IL-9 were used following published protocols [38,41], while TGF-β3 ELISA was performed according to a protocol established at the Ludwig Institute for Cancer Research Brussels branch in collaboration with Dr. Catherine Uyttenhove [14] (Appendix A). Briefly, plates were coated with MTGF-β3-17 antibody at 4 µg/mL overnight and saturated with 10% FCS. Then, 1:2 diluted plasma samples were added and incubated at 37 °C for 2 h. Biotinylated antibody MGFT-β3-16 was added and incubated at 37 °C for 1 h and 30 min. Avidin HRP was added and incubated at 37 °C for 1 h. The detection was evaluated using TMB at 37 °C [14] (Appendix A shows both TGF-β1 and TGF-β3 ELISA specificity and sensitivity). For the detection of latent TGF-β1 and β3 (since no active TGF-β was detected), the samples (50 µL) were incubated with 10 µL 1N HCL for 10 min at room temperature before the addition of 10 µL 1.2 N NaOH/Hepes 0.5 N. All cytokine levels were measured in duplicate, and each plate included a standard curve of known positive concentrations and negative control. Absorbance was read against a blank at 450 nm using an ELISA reader.

### 4.4. Evaluation of FOXP3 Expression via Quantitative RT-PCR

The Ficoll–Paque gradient method was used for the isolation of 324 PBMC samples, as described previously [42]. The isolation of total RNA from PBMCs was performed with an RNeasy Mini Kit (QIAGEN, Hilden, Germany), according to the manufacturer’s instructions. The amount of FOXP3 was detected using a qRT-PCR assay on a 7500 Fast Real-time PCR system (Applied Biosystems). Briefly, 20 μL cDNA was synthesised from 2 μg RNA using Oligo (dT)_15_ primer 200 pmol/µL. Then, 3 μL of cDNA was mixed with 10 μL SYBR Green qPCR master mix, and 1 μL FOXP3 Primer–Probe (5′-AACTATGAAACAAATTTTCCT-3′ and 5′-TTAGGAAAATTTGTTTCATAG-3′) was run in the final volume of 20 μL, with GADPH (5′-GTCTCCTCGATTCAACAGCG-3′ and 5′-ACCACCCTGTTGCTGTACCAA-3′) as the internal control. The programme was set up following a constant protocol: the first step at 50 °C for 2 min, initial heating at 90 °C for 10 min, 50 cycles of denaturation at 95 °C for 15 s, annealing at 60 °C for 1 min, and elongation at 60 °C for 1 min. The samples were assayed in duplicate, and the average cycle threshold (ΔCT-FOXP3) was calculated, while GAPDH was used as the internal control housekeeping gene. The relative fold expression changes in the FOXP3 gene compared with the internal control were determined.

### 4.5. Statistical Analysis

The results of this study were statistically analysed using the GraphPad Prism Version 9.0.0 (121) programme and R statistics version 4.1.3. The mean comparisons between the two groups were performed using a *t*-test, while one-way analysis of variance (ANOVA) was used for mean comparisons between multiple groups, followed by a post hoc Turkey analysis for pairwise contrasts. In addition, factor analysis was used for dimension reduction and to determine the contribution of different parameters to clinical outcomes. All ELISA data were log-transformed, and a *p*-value less than 0.05 (*p* ≤ 0.05) was considered statistically significant.

## Figures and Tables

**Figure 1 ijms-23-12665-f001:**
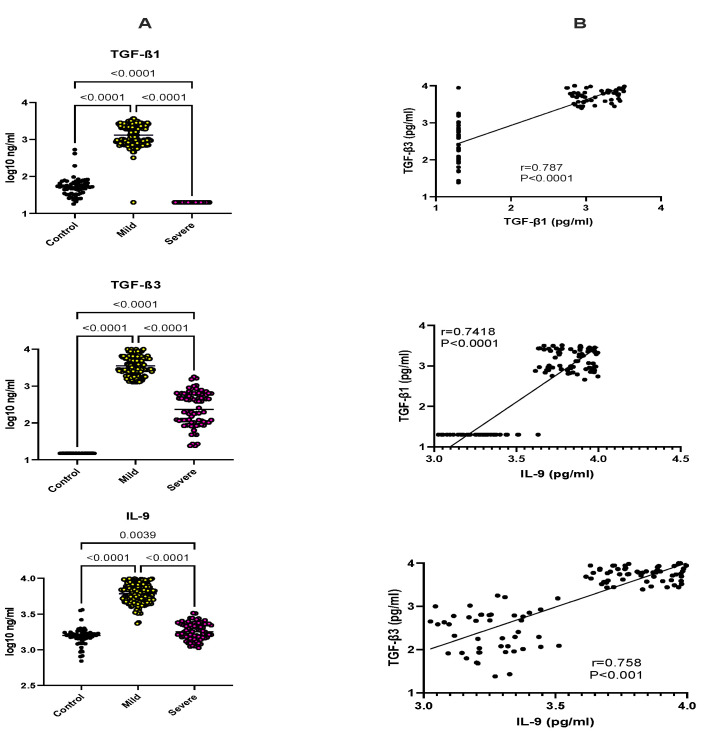
(**A**) TGF-β1, TGF-β3, and IL-9 levels in malaria patients. Plasma samples from a panel of control donors and mild or severe malaria patients were tested for TGF-β1, TGF-β3, and IL-9 by using ELISA. For TGF-β1 and TGF-β3 assays, samples were briefly exposed to 0.1 M acetic acid (Ph 3), followed by TRIS buffer (Ph8), before performing the ELISA. Statistical significance was calculated by one-way ANOVA; (**B**) correlation of TGF-β1, TGF-β3, and IL-9 plasma concentrations in individual mild and severe patients. Spearman r correlation coefficients are indicated.

**Figure 2 ijms-23-12665-f002:**
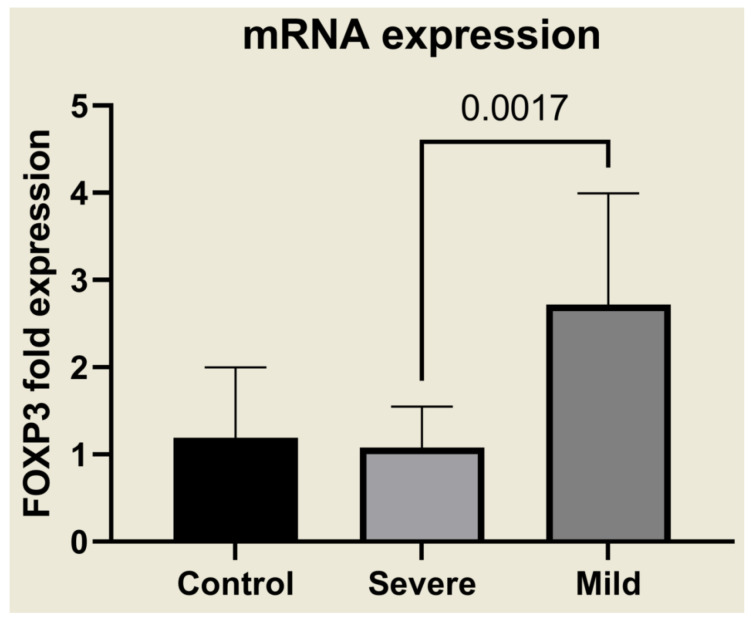
FOXP3 expression in malaria patients. mRNA extracted from PBMCs was tested for FOXP3 with a Qrt-PCR assay in mild and severe cases of malaria.

**Figure 3 ijms-23-12665-f003:**
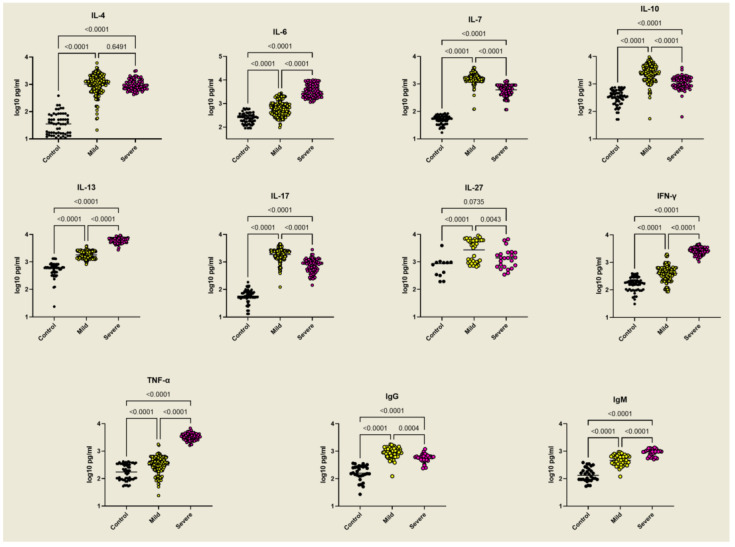
Cytokine and Immunoglobulin levels in mild and severe malaria patients and controls. A panel of control donors and mild or severe malaria patients was tested for the indicated cytokines, IgG, and IgM by using ELISA. Each dot corresponds to an individual patient. The significance of the differences was calculated by one-way ANOVA (*p* ≤ 0.05, *p* ≤ 0.001, and *p* ≤ 0.0001).

**Figure 4 ijms-23-12665-f004:**
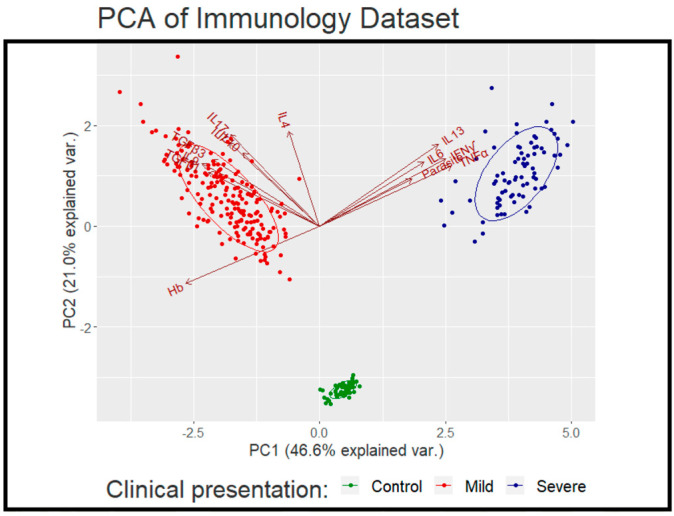
Dimensional analysis indicated the strongest contribution of TNF-α, IFN-γ, and IL-13 towards disease severity, while parasitaemia had the weakest contribution. On the other hand, TGF-β1, TGF-β3, and IL-9 had the strongest contribution towards mild cases. The levels of IL-4 were not significantly different between mild and severe cases.

**Figure 5 ijms-23-12665-f005:**
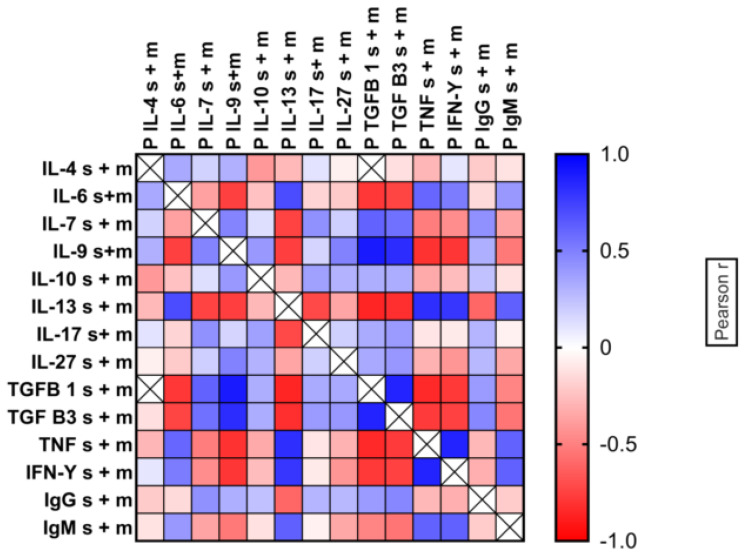
Heat map correlations calculated by Pearson protocols for all cytokines in combined mild and severe malaria patients.

**Table 1 ijms-23-12665-t001:** Diversity of Plasmodium species present in each group of participants.

Plasmodium Species Identification	% Mild Case; *n* = 186	% Severe Anaemic Malaria, *n* = 80
*Plasmodium falciparum*	82.79	88.75
*Plasmodium ovale*	6.45	2.5
*Plasmodium malariae*	6.82	7.5
*Plasmodium Vivax*	0.53	0
Mixed		
Plasmodium infections		
*P. falciparum* and *P. malariae*	1.61	1.25
*P. falciparum* and *P. ovale*	1.61	0

## Data Availability

The datasets presented in the current study are available from the corresponding author upon reasonable request.

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
