# Peer review of "Integrated Analysis of Cytokine Profiles in Malaria Patients Discloses Selective Upregulation of TGF-β1, β3, and IL-9 in Mild Clinical Presentation"

_ijms, 2022, doi:10.3390/ijms232012665_

Round 1

Reviewer 1 Report

There are obvious typographical errors in the study.

The lack of malaria species determination was a major shortcoming.

It is not enough to make the diagnosis only with a microscope. The diagnosis should be confirmed with PCR.

Without these, the work is not of sufficient originality.

Author Response

There are obvious typographical errors in the study.

Response: The manuscript has been thoroughly reviewed for typographical errors

The lack of malaria species determination was a major shortcoming.

Response: Malaria species distribution for the 2 groups of patients has now been reported (supplementary table 2).

It is not enough to make the diagnosis only with a microscope. The diagnosis should be confirmed with PCR.

Response: We thank the reviewer for this comment. However, although PCR is a more sensitive diagnostic technique, microscopy is a widely accepted malaria diagnostic tool and one that is recommended by the WHO as a diagnostic standard to guide treatment strategies (https://www.who.int/teams/global-malaria-programme/case-management/diagnosis/microscopy). In our study, for quality assurance, all slides read by a health care technician were validated by a highly trained microscopist from the national reference laboratory.

Reviewer 2 Report

Ndoricyimpaye and collaborators described in their manuscript an Integrated analysis of cytokine profiles in malaria patients. A total of 324 participants from Ruanda were enrolled allocated in three groups: 80 with severe anemic malaria,186 with a mild clinical presentation and a group of 58 non-infected controls.

The authors measured the levels of a range of cytokines and correlated with TGF-β1 and TGF-β3 production.The obtained data suggest unique association between IL-9 and FOXP3 induced TGF-ß1 and TGF-ß3 that is a strong predictor for a mild clinical presentation in human natural malaria infection. Overall, the manuscript is well written and organized. The experiments were well designed and the results were well described and discussed. This manuscript deserves be accepted for publication after minor adjustments. For this reviewer, the authors should include more details in the abstract such as  the number of enrolled individuals and local. 

Author Response

reviewer 2:

Ndoricyimpaye and collaborators described in their manuscript an Integrated analysis of cytokine profiles in malaria patients. A total of 324 participants from Ruanda were enrolled allocated in three groups: 80 with severe anemic malaria,186 with a mild clinical presentation and a group of 58 non-infected controls.

The authors measured the levels of a range of cytokines and correlated with TGF-β1 and TGF-β3 production. The obtained data suggest unique association between IL-9 and FOXP3 induced TGF-ß1 and TGF-ß3 that is a strong predictor for a mild clinical presentation in human natural malaria infection. Overall, the manuscript is well written and organized. The experiments were well designed and the results were well described and discussed. This manuscript deserves be accepted for publication after minor adjustments. For this reviewer, the authors should include more details in the abstract such as  the number of enrolled individuals and local.

We thank the reviewer for the positive feedback and we have added details in the abstract as requested

Round 2

Reviewer 1 Report

The authors have fixed the word and symbol errors.

It has been shown that the species determination is made in  Supplementary Table 2. However, when I open the file, the type assignment does not appear in table 2.  There are articles about induced different levels of IL between Plasmodium species (Goncalves, R. M., Scopel, K. K., Bastos, M. S., & Ferreira, M. U. (2012)). Cytokine balance in human malaria: does Plasmodium vivax elicit more inflammatory responses than Plasmodium falciparum?) In addition, Depending on the immune system status, the characteristics of the people in the region or idiopathic, there may be P.falciparum resistant people, as well as those with severe clinical symptoms of P.vivax infection. For this reason, the results should be interpreted separately by determining the species.

Author Response

It has been shown that the species determination is made in Supplementary Table 2. However, when I open the file, the type assignment does not appear in table 2.  There are articles about induced different levels of IL between Plasmodium species (Goncalves, R. M., Scopel, K. K., Bastos, M. S., & Ferreira, M. U. (2012)). Cytokine balance in human malaria: does Plasmodium vivax elicit more inflammatory responses than Plasmodium falciparum?) In addition, Depending on the immune system status, the characteristics of the people in the region or idiopathic, there may be P.falciparum resistant people, as well as those with severe clinical symptoms of P.vivax infection. For this reason, the results should be interpreted separately by determining the species.

Response:

We thank the reviewer for these comments. The table that shows the determined species have been added to the results. A single P. vivax, 14 P.malariae, 15 P.ovale and 6 mixed plasmodium species infections were identified in the present study. However, we could not detect any difference in immune responses between P.ovale, P. malariae and P.falciparum infected people. This may have been due to the low level of non-falciparum infections and this has been discussed in the revised MS.

Round 3

Reviewer 1 Report

It is appropriate to publish this version of the study.

Author Response

                Thank You for the comment, minor spell checks have been corrected in the text.